# Generation of photonic entanglement in green fluorescent proteins

Siyuan Shi[1], Prem Kumar[1,2] & Kim Fook Lee[2]

Recent development of spectroscopic techniques based on quantum states of light can precipitate many breakthroughs in observing and controlling light-matter interactions in biological materials on a fundamental quantum level. For this reason, the generation of entangled light in biologically produced fluorescent proteins would be promising because of their biocompatibility. Here we demonstrate the generation of polarization-entangled two-photon state through spontaneous four-wave mixing in enhanced green fluorescent proteins. The reconstructed density matrix indicates that the entangled state is subject to decoherence originating from two-photon absorption. However, the prepared state is less sensitive to environmental decoherence because of the protective β-barrel structure that encapsulates the fluorophore in the protein. We further explore the quantumness, including classical and quantum correlations, of the state in the decoherence environment. Our method for photonic entanglement generation may have potential for developing quantum spectroscopic techniques and quantum-enhanced measurements in biological materials.

[1] Department of Physics and Astronomy, Center for Photonic Communication and Computing, Northwestern University, 2145 Sheridan Road, Evanston, IL 60208-3112, USA. [2] Department of Electrical Engineering and Computer Science, Center for Photonic Communication and Computing, Northwestern University, 2145 Sheridan Road, Evanston, IL 60208-3118, USA. Correspondence and requests for materials should be addressed to S.S. (email: siyuanshinu@gmail.com) or to P.K. (email: kumarp@northwestern.edu) or to K.F.L. (email: kflee70@gmail.com)

Quantum spectroscopy exploits the quantum nature of light for exploring light-matter interactions[1]. The distinctive advantages of quantum light in spectroscopy stem from improved signal-to-noise ratio owing to strong correlation between light and matter, particularly quantum entanglement[2]. Entangled light beams retain an inherent feature that the quantum state of each beam cannot be described independently of the others. Specifically, entangled photon pairs promise to enhance the precision of measurements[3, 4], as well as provide novel control knobs in nonlinear spectroscopic applications[5, 6]. Advances in quantum biology[7] have driven our attention to novel quantum spectroscopic techniques in biological materials where the entangled light is generated in situ.

Fluorescent proteins (FPs) have received significant attention in biomedical research (fluorescence microscopy[8], intracellular dynamics[9], and reporter gene technology[10]) because of their high quantum efficiency in absorption-emission processes[11, 12] and their ability to fuse to other proteins while maintaining fluorescence[13]. The optofluidic biolaser[14], where the gain medium consists of enhanced green FP (EGFP) expressed in live cells (the definition of EGFP is given in Methods), has succeeded in measuring subtle changes in biological molecules. The potential application of FPs in quantum technology, however, still awaits exploration. The generation of nonclassical light, such as squeezed and entangled light, in biologically produced FPs would open up the potential for quantum spectroscopy and quantum-enhanced measurements in biological systems because entanglement can provide precision that surpasses the uncertainty principle[4]. Entanglement can be generated in FPs owing to the strong optical nonlinearity[15] and the process can be quite efficient because of the protective β-barrel structure surrounding the fluorophore in the protein[16]. In addition, since the FPs strongly couple to light, even at the single-photon level, it is feasible that quantum-optic techniques for creating, manipulating, and characterizing photonic quantum states, developed for use in quantum information processing, could be directly applied to FPs. By projecting the entanglement of the quantum fields back to FPs, precise preparation and control of higher excited states[17–19] may be feasible.

In this work, we generate polarization-entangled two-photon state in an ensemble of fluorophores in EGFP through spontaneous four-wave mixing (SFWM), where the energy levels of the ground and excited states are nearly two-photon resonant with the pump light (the SFWM process in EGFP is described in Methods). The generation of an entangled photonic state indicates the preservation of quantum superposition and coherence within the ensemble of excited fluorophores in two different time slots and polarizations. We verify the polarization correlation of the state by performing two-photon interference (TPI) measurements. Moreover, we characterize the state by means of quantum-state tomography (QST). We then explore the quantumness, including entanglement and classical and quantum correlations, of the state in the decoherence environment. Our study suggests that two-photon absorption (TPA) reduces the entanglement and other quantum features of the state. Our experiments show that the biologically produced EGFP can serve as a promising entangled photon-pair source despite the protein molecules being in strongly decohering environment of a room-temperature solution. This is due to the encapsulating β-barrel structure that protects the fluorophores. What's more, since EGFP can be genetically engineered and expressed in biological cells, our results may have the potential for realizing practical biomimetic quantum networks[20] and quantum sensors[21].

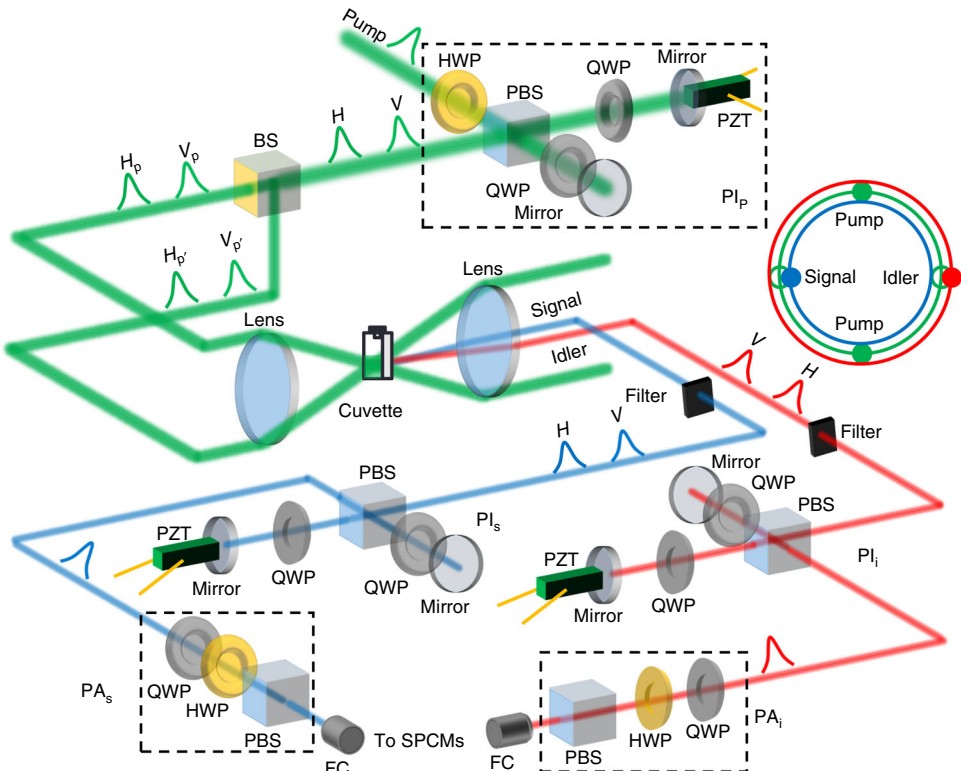

**Fig. 1** Experimental setup. The unbalanced polarization interferometer ($PI_{p,s,i}$) contains a half-waveplate (HWP), a polarization beam splitter (PBS), two quarter-wave plates (QWP), and two mirrors with one mounted on a piezoelectric transducer (PZT). The polarization analyzer in the signal/idler ($PA_s$/$PA_i$) channel contains a QWP, a HWP, and a PBS. Inset: the phase-matching cone for nondegenerate signal and idler photons. The two green spots represent the two pump beams. The blue and red spots represent our selected directions for detecting the entangled signal and idler photons

## Results

**Generation of polarization-entangled photon pairs.** In the experiment, we generate horizontally ($|H_sH_i\rangle$) and vertically ($|V_sV_i\rangle$) polarized photon-pair amplitudes in two time slots through SFWM and then combine them using polarization interferometers to form a polarization-entangled two-photon state $(|H_sH_i\rangle + e^{i2\phi_p}|V_sV_i\rangle)/\sqrt{2}$, where the pump phase ($\phi_p$) is carefully controlled. The experimental setup for generating and characterizing the entangled photonic state is shown in Fig. 1. The EGFP is kept in a 5 mm-long cuvette. According to the instructions provided by the vendor, our EGFP is prepared with a molar concentration of 25.5 µM in phosphate buffered saline solution. The pump pulses are obtained from a mode-locked regenerative amplifier (Coherent Inc., RegA-9000, seeded by Mira-900 and pumped by Verdi-10) emitting pulses with a duration of $\tau_p = 200$ fs at the center wavelength of 785 nm. In our experiment, the SFWM process requires high-peak pump power that may cause photo-induced damage and photobleaching (photochemical alteration of the fluorophores) in the EGFP[22, 23]. The irreversible photobleaching effect can reduce the fluorescence efficiency and the brightness of photons generated via the SFWM process. To keep photobleaching at a negligible level[22], we use low-repetition-rate (40 kHz) pump pulses; only at repetition rates of 80 kHz or higher do we see evidence of photobleaching in our experiment.

The pump pulses are split into horizontally and vertically polarized components by using an unbalanced polarization interferometer ($PI_p$) with a temporal delay of $\Delta t = 33.3$ ps, which can be tuned by a piezoelectric transducer (PZT). The pump pulses are then split again by using a 50:50 beam splitter (BS) for creating two horizontally ($H_pH_{p'}$) and vertically ($V_pV_{p'}$) polarized components in the two time slots, where p and p' indicate pumps along the top and bottom paths, respectively. The four pump pulses are then focused into the cuvette by using a lens with a focal length of 50 cm. The beam waists are about 40 µm in diameter, corresponding to a confocal parameter (twice the Rayleigh range) of 13.9 mm. The signal and idler beams lie on a phase-matching cone, whose cross-section is shown in the inset of Fig. 1. Spatial overlapping of the pump pulses ($H_p$ and $H_{p'}$, $V_p$ and $V_{p'}$) is very critical for generating a maximally polarization-entangled state. Slight misalignment can completely destroy the entanglement in the detected photon pairs because of the non-overlapping spread of the photon wave functions along the measured directions[24, 25]. We can avoid this spatial decoherence by ensuring the same FWM gain for the four combinations of the two pump pulses, viz, $H_pH_{p'}$, $V_pV_{p'}$, $H_pV_{p'} \rightarrow H_pH_{p'}$, and $V_pH_{p'} \rightarrow V_pV_{p'}$, where the last two combinations are obtained by rotating $H_{p'} \leftrightarrow V_{p'}$ and interchanging time slots 1↔2. The transverse spatial profiles of the pump pulses are quantified by coupling them into single-mode fibers with efficiencies >70% (not shown). The $M^2$ factor of the beam quality is < 2.

Behind the sample, we use a notch filter with a bandwidth of 33 nm at the center wavelength of 785 nm for blocking the scattered pump photons in the signal and idler directions (channels). We select the signal and idler photons at the center wavelengths of $\lambda_s = 730$ nm and $\lambda_i = 849$ nm, respectively. The spectral isolation is obtained by using tunable bandpass filters with bandwidths of 20 nm and transmission efficiencies of 98% as shown in Supplementary Fig. 1. The filters provide an isolation of >100 dB from the pump photons.

Unbalanced polarization interferometers in the signal ($PI_s$) and idler ($PI_i$) arms are used to compensate for the time delay that is created by $PI_p$ to form the entangled photonic state $(|H_sH_i\rangle + e^{i2\phi_p}|V_sV_i\rangle)/\sqrt{2}$. The polarization analyzer (PA), which consists of a quarter waveplate, a half waveplate, and a polarizing BS, is used to characterize the quantum state. The free-

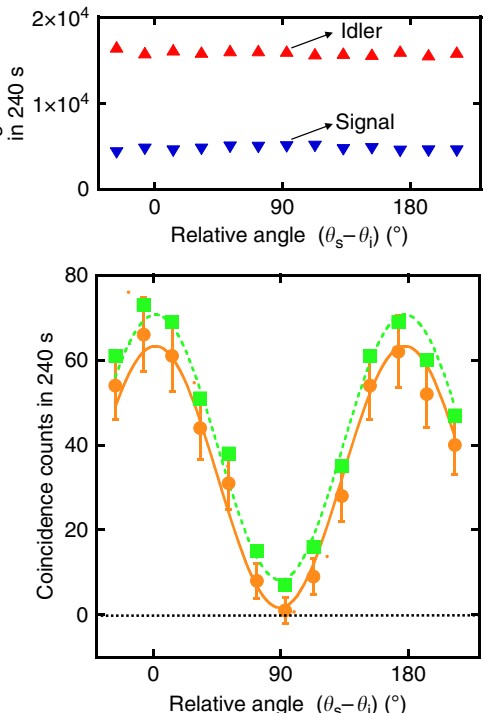

**Fig. 2** Quantum interference of entangled photon pairs. Top: photon counts in the signal and idler channels as the relative polarization angle between the signal and idler ($\theta_s - \theta_i$) is varied. Red upward-pointing triangles and blue downward-pointing triangles represent photon counts in the idler and signal channels, respectively. Bottom: variation in coincidence counts (CCs) vs. the relative polarization angle ($\theta_s - \theta_i$). Orange circles and green squares represent CCs with and without subtraction of accidental-coincidence counts (ACs). The solid orange curve and the dashed green curve are sinusoidal fits to the data. Each data point is recorded for an integration time of 240 s, while the pump phase $\phi_p$ is fixed at 0°. The error bars are calculated based on the standard deviation of photon counts using Poisson statistics

space fiber collimators collect the signal and idler photons and send them to the single-photon counting modules (SPCMs). The dark-count probability of the SPCMs is about $10^{-5}$ per pulse. The total detection efficiency for the signal (idler) photons is 60% (50%). A correlator (CPDS, NuCrypt LLC) measures the correlation of the counts from both the SPCMs. In our detection system, a coincidence count (CC) is recorded when both SPCMs detect a photon in the same gated time interval, while an accidental-coincidence count (AC) is recorded when both SPCMs detect a photon in the adjacent gated time intervals.

**Two-photon interference.** In our experiment, we record the TPI pattern by varying the relative polarization angle between $PA_s$ ($\theta_s$) and $PA_i$ ($\theta_i$). The interference is visualized as a sinusoidal behavior of the CCs vs. the difference between $\theta_s$ and $\theta_i$. Figure 2 shows the TPI fringe. With an integration time of 4 min for each data point during which $\phi_p$ is kept fixed, we obtain a fringe visibility of 98% (85%) with (without) subtraction of the ACs. The relative phase between the $PI_s$ and $PI_i$ is stable for more than half an hour. We optimize the interferometric visibility of $PI_s$ and $PI_i$ by guiding the pump beam into the signal and idler paths. These individual interferometric visibilities set an upper limit on the measurable TPI visibility of 98% in our experiment, in agreement with the AC-subtracted value above. Since the photon pairs are generated in an ensemble of excited fluorophores at two different time slots, the probability amplitudes of $|H_sH_i\rangle$ and $|V_sV_i\rangle$ are

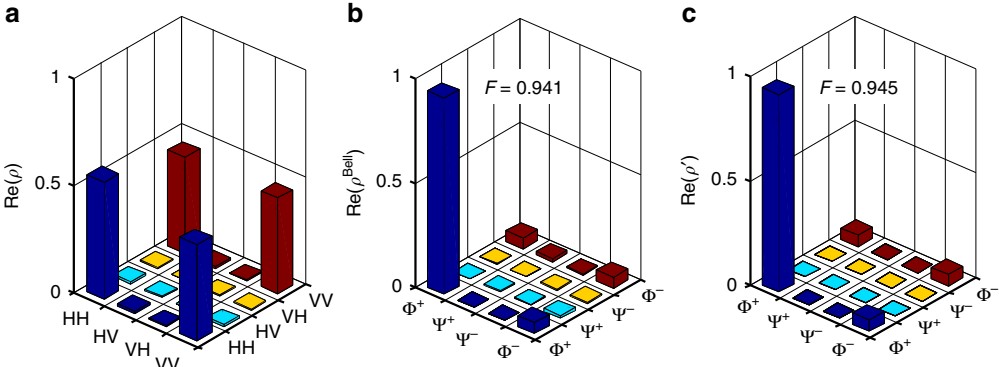

**Fig. 3** Measured density matrix of the entangled photonic state. **a** The reconstructed state in the horizontal–vertical basis. **b** The reconstructed state in the Bell-state basis. **c** The estimated state in the Bell-state basis. $\Phi^+ = (|H_sH_i\rangle + |V_sV_i\rangle)/\sqrt{2}$, $\Psi^+ = (|H_sV_i\rangle + |V_sH_i\rangle)/\sqrt{2}$, $\Psi^- = (|H_sV_i\rangle - |V_sH_i\rangle)/\sqrt{2}$, and $\Phi^- = (|H_sH_i\rangle - |V_sV_i\rangle)/\sqrt{2}$ are the four Bell states. Only the real components are shown, the imaginary components are all less than few percent

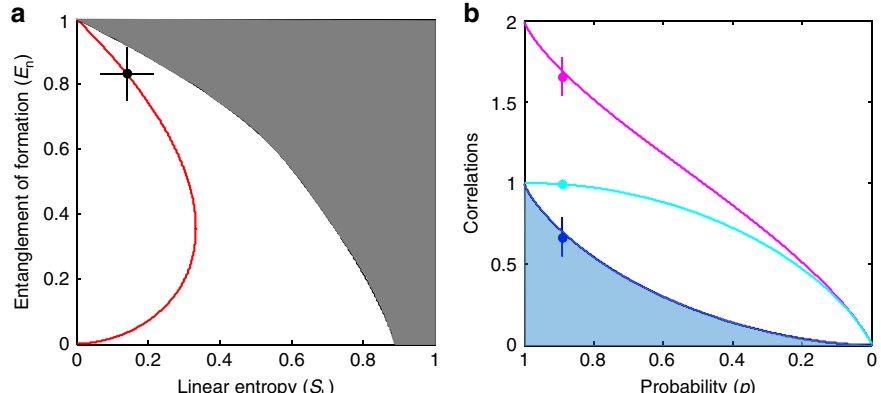

**Fig. 4** Entanglement of formation, linear entropy, and correlations of the state. **a** Entanglement of formation ($E_n$) vs. linear entropy ($S_L$). The solid circle represents the experimentally generated entangled photonic state. The solid red curve represents the behavior of $\rho'(p)$ on the plane. The gray region represents the physically impossible combinations of $E_n$ and $S_L$. **b** The total correlation, classical correlation, and quantum correlation vs. probability $p$. The solid magenta, cyan, and blue circles represent total correlation ($I$), classical correlation ($C$), and quantum correlation ($Q$) of the experimentally generated entangled photonic state. The solid magenta, cyan, and blue curves represent the theoretical behaviors of $I$, $C$, and $Q$ as functions of $\rho'(p)$. The curved edge of the light blue region represents the relative entropy ($R_n$) of $\rho'(p)$. The error bars are calculated based on the standard deviation of photon counts using Poisson statistics

subject to the dynamic decoherence environment inside the EGFP.

**Quantum state of the photon pairs**. We prepare the Bell state $|\Phi^+\rangle = (|H_sH_i\rangle + |V_sV_i\rangle)/\sqrt{2}$ by setting $\phi_p = 0$. We then characterize the state by employing the standard method of QST using 16 measurement settings in different basis. Each setting is obtained by adjusting the $PA_s$ and $PA_i$ to provide a glimpse of a distinct aspect of the quantum state's reality. For each setting, the CCs and ACs are collected with an integration time of 2 min. A maximum-likelihood estimation algorithm reconstructs the $4 \times 4$ density matrix ($\rho$) of our state with a fidelity $\left( F = \sqrt{\langle \Phi^+ | \rho | \Phi^+ \rangle} \right)$ of $0.94 \pm 0.05$ in the horizontal–vertical (HV) basis as shown in Fig. 3a.

The measured density matrix unveils many aspects of the experiment that involve both the quality of the experimental setup and the effects of decoherence inside the EGFP. The small imaginary components indicate that the $PI_s$ and $PI_i$ are stable, without introducing a phase drift between the $|H_sH_i\rangle$ and $|V_sV_i\rangle$ amplitudes. The four distinct peaks with values satisfying $\rho_{HHHH} > \rho_{HHVV} = \rho_{VVHH} \approx \rho_{VVVV}$ suggest that the experiment generates a mixed state. It's clearer to view the matrix on the Bell-state basis

($\rho^{Bell}$) as shown in Fig. 3b, where $\rho^{Bell}_{\Phi^+\Phi^+} = 0.941$ is the dominating component and $\rho^{Bell}_{\Phi^+\Phi^-} = \rho^{Bell}_{\Phi^-\Phi^+} \approx \rho^{Bell}_{\Phi^-\Phi^-} = 0.052$. By neglecting the other (non-corner) much smaller components, we approximate the density matrix as $\rho'(p) = p|\Phi^+\rangle\langle\Phi^+| + (1-p)|H_sH_i\rangle\langle H_sH_i|$, where the probability $p = 0.89$. The fidelity of $\rho'(0.89)$ is 0.945, close to the fidelity of $\rho$, as shown in Fig. 3b, c, respectively. Another two QST measurements with integration times of 4 and 6 min (for each setting) also result in similar mixed states as discussed in Supplementary Fig. 2. Details on the origin of this decoherence effect are later given in the discussion.

**Quantumness of the state**. The degree of entanglement, as well as the degree of purity of a quantum system[24] are crucial criteria for realizing quantum information protocols in a decohering environment. The entanglement of formation ($E_n$)[26] can quantify the degree of entanglement in an arbitrary two-qubit system and the linear entropy ($S_L$)[27] can quantify the degree of purity (the analytical expressions for $E_n$ and $S_L$ of $\rho'(p)$ can be found in Methods). We place our measured state and $\rho'(p)$ on a characteristic plane constructed by $E_n$ and $S_L$, as shown in Fig. 4a. A pure unentangled state lies at (0, 0); a pure maximally entangled state lies at (0, 1); a maximally mixed and unentangled state lies at

(1, 0). The boundary between the white and gray regions represents the maximally entangled mixed states[25]. $\rho'(p)$ (red solid curve) varies along the curve from that for a pure maximally entangled state (0, 1) to that for a pure unentangled state (0, 0) as $p$ decreases from 1 to 0. Our experimentally generated state (black circle) lies on this red curve at (0.14, 0.83).

It is important to distinguish the quantum correlation from the total correlation of a quantum system in a decoherence environment because some quantum tasks may not utilize the entanglement but still exhibit quantum advantages[28, 29]. For a bipartite system consisting of two subsystems A and B, the quantum mutual information ($I$)[30], which is a measure of the total correlation between subsystems A and B, is given as $I(\rho_{A:B}) \equiv S(\rho_A) + S(\rho_B) - S(\rho_{AB})$, where $S(\rho) = -\text{tr}[\rho \log_2 \rho]$ is the Von Neumann entropy. The classical correlation ($C$)[31] is widely accepted as $C(\rho_{AB}) \equiv \max_{B^\dagger B}[S(\rho_A) - \sum_i p_i S(\rho_A^i)]$. In this definition, $B^\dagger B$ is a positive-operator-valued measure performed on the subsystem B and $\rho_A^i = \text{tr}_B(B_i \rho_{AB} B_i^\dagger)/\text{tr}_{AB}(B_i \rho_{AB} B_i^\dagger)$ is the post-measurement state of the subsystem A after obtaining the outcome $i$ with probability $p_i$ on the subsystem B. The quantum correlation ($Q$), which quantifies the correlation that cannot exist in any classical state, is given by $I - C$. In Fig. 4b, we show the correlations ($I$, $C$, $Q$) for our generated state and for $\rho'(p)$ in general (the analytical expressions for the correlations of $\rho'(p)$ are given in the Methods section). For $\rho'(p)$ (solid curves), $I$, $C$, and $Q$, all decay monotonically to 0 at $p = 0$. The $I$ and $Q$ of our generated state (solid circles) are slightly off the curves because we have neglected the small non-corner components in the measured density matrix. Relative entropy of entanglement ($R_n$)[32] can quantify the distance between entanglement and quantum correlation. In Fig. 4b, $R_n$ of $\rho'(p)$ (the curved edge of the light blue region) overlaps with $Q$ (the analytical expression for $R_n$ is given in Methods). It means that the entanglement is the only contributing factor to the quantum correlation in $\rho'(p)$. Therefore, we conclude that the decoherence effect in EGFP can induce an additional component ($|H_s H_i\rangle\langle H_s H_i|$) in the prepared state. The decoherence effect can reduce the classical correlation, the quantum correlation, and, therefore, the total correlation. In the extreme case when $p = 0$, the state is a pure unentangled state without any correlations. Moreover, the decoherence effect cannot induce any non-entangled quantum correlations.

## Discussion

The decoherence effect originates from the nonlinear optical processes occuring inside the EGFP. The dominating process is TPA[33]. In TPA, two pump photons can excite the electronic system from the ground state ($S_0$) to an excited vibronic state in the $S_1$ manifold due to enhancement of certain vibronic transitions[15]. Then the system decays (on time scale of a few picoseconds) to the lowest vibronic state in $S_1$ through non-radiative relaxation. The lifetime of $S_1$ is ~3 ns[34]. A fluorescence photon is emitted simultaneously with the spontaneous transition from $S_1$ to $S_0$. In our case, the probability that a fluorophore undergoes TPA is ~0.1 within one pump-pulse duration. In the experiment, two horizontally (vertically) polarized pump pulses generate horizontally (vertically) polarized photon pairs in an ensemble of EGFP molecules through the SFWM process, while simultaneously, the pulses can pump the molecules to $S_1$ via TPA with a probability of ~0.1. Note that the temporal delay of the orthogonally polarized pump pulses ($\Delta t = 33.3$ ps) is much less than the lifetime of $S_1$ (~3 ns). When the vertically polarized pump pulses enter the sample, ~10% of the molecules (which have absorbed two photons) are in $S_1$ and hence cannot contribute to the production of photon pairs through SFWM. What is more

important is that the molecules that generated the quantum amplitude for horizontally polarized photon pairs may absorb two vertically polarized pump photons and thus be unable to produce the quantum amplitude for vertically polarized photon pairs. The net outcome is the emergence of an additional component ($|H_s H_i\rangle\langle H_s H_i|$) in the prepared two-photon state. This effect certainly reduces the entanglement and the quantum correlation of the generated photons. Since the frequency-dependent lineshape functions of $\sigma_2$ (TPA cross-section) and $n_2$ (nonlinear refractive index) are different in a two-level system with permanent dipole moments[35], a practical way to increase the entanglement is by choosing the pump wavelength such that $\sigma_2$ is relatively small, whereas $n_2$ is near its peak.

It is assumed that the biological molecules, which exist close to room temperature and in solution, are continuously interacting with their environments. As a result, the molecular superpositions are assumed to be inhibited by the continuous quantum measurements and the associated wave-function collapse. However, we see from our experiment that the EGFP in solution is resistant to such continuous environmental decoherence. We attribute this to the protective β-barrel structure of the EGFP that encapsulates the fluorophore. The structure not only protects the fluorophore from contact with the surrounding solvent, but also manifests strong stability to thermal and chemical denaturation by multiple noncovalent interactions[36].

Since the entangled photon pair is better preserved than the correlated photon pair in multiple scattering media[37], our entangled photon pairs can be used as a heralded single-photon source for biomedical imaging. In addition, the TPI visibility of ~85% (~98%) for the entangled photon pairs can provide signal-to-noise ratio as high as 10 dB (20 dB) for sensing and imaging through the coincidence basis. Therefore, we can use the CCs of the photon pairs for observing spatiotemporal dynamics of proteins with resolution surpassing the diffraction limit[19, 38]. The spectral[39] and spatial properties[40] of the polarization-entangled photon pairs can also be used for performing bi-photon spectroscopy[41] and coincidence imaging (ghost imaging)[42] such as measuring the spectral and spatial properties of the EGFP-expressing cells[43] via coincidence basis measurements, respectively. For example, we can design the wavelength range for the signal and idler such that the signal photons travel through the cells under study experiencing phase change and loss while the idler photons travel through the cells without disturbance. The signal photons are detected by using a single-photon detector. As for the idler photons, we insert a tunable filter or a lens before a single-photon detector for performing the local spectral or spatial analysis, respectively. We can then extract out the spectral and spatial properties of the cells under study via coincidence basis measurements. We can choose the polarization projection angles of the signal and idler photons to be 45° for the Bell state $|\Phi^+\rangle$ as we perform the coincidence imaging or bi-photon spectroscopy. The cells under study can be similar to the transfected mammalian cells (293ETN cells derived from the human embryonic kidney cell line HEK293) with a plasmid encoding for EGFP[43]. Moreover, the entangled photon pairs can be engineered for manipulating the vibronic states in FPs through two-photon excitation. Their superiority originates from the simultaneous absorption of the entangled photons[19], thus avoiding the decay process in the intermediate states that occur when using classical light.

Even more intriguing, however, is the possibility of developing an experimental heuristic for quantum effects in EGFP. Since EGFP can be expressed in living cells, the genetic sequences encoding the residues that define their structures and physical characteristics can be altered. There is a major advantage to using a biological system for generating the fluorophores of interest:

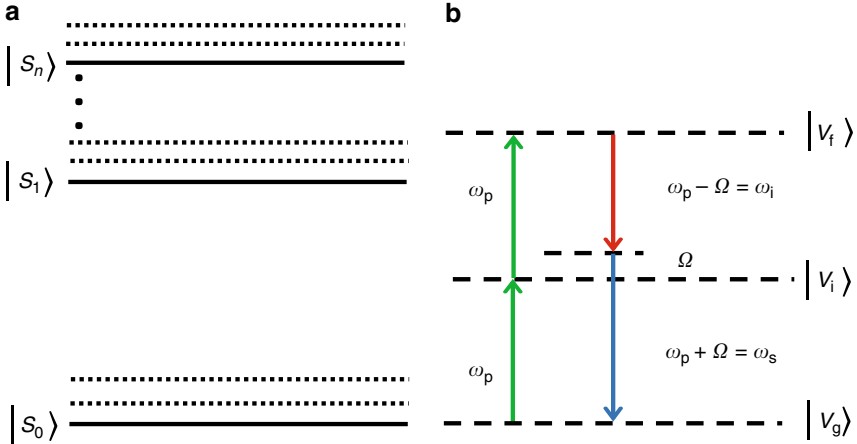

**Fig. 5** Partial energy-level diagram of EGFP and the virtual levels involved in the SFWM process. **a** The relevant partial energy-level diagram of EGFP. The solid lines represent energy levels including the ground state ($S_0$), the first excited state ($S_1$), and the higher excited states ($S_n$). The dotted lines represent vibronic levels. **b** Virtual levels involved in the spontaneous four-wave mixing (SFWM) process. The dashed lines represent the virtual states ($V_g$, $V_i$, and $V_f$) that are involved in the SFWM process. $\omega_p$, $\omega_s$, and $\omega_i$ are center frequencies of the pump, signal, and idler fields, respectively, and $\Omega$ is the frequency detuning

physical characteristics can be rapidly and efficiently optimized via random mutagenesis[9], and thus enabling a process by which entanglement sources can be genetically engineered. For this reason, our FP-based entangled photon source is bio-compatible and comparable to quantum dots as fluorescent labels in medical applications. In addition, our photonic entanglement generation scheme in EGFP can be easily extended to high dimensions ($d > 2$) such as for generating a time-bin polarization-entangled state. Also, the observed fidelity and quantum correlation of the entangled photon pairs generated in EGFP encourage us to apply the quantum illumination technique[44] in EGFP-expressing cells in the future.

In conclusion, we have generated polarization-entangled two-photon states with high fidelity through the SFWM process in EGFP. The measured density matrix unveils the fidelity-limiting decoherence effect that originates from TPA inside the EGFP. Moreover, our prepared state is free from environmental decoherence because of the protective β-barrel structure that encapsulates the fluorophore in the protein. Our photonic entanglement generation and characterization indicate that the SFWM process in EGFP is a promising quantum process for developing quantum spectroscopic techniques and quantum-enhanced measurements in biological materials.

## Methods

**EGFP**. The EGFP is a mutant of the original wild-type GFP. It has double substitutions of the amino acids: S65T and F64L. S65T is considered essential for suppressing the 395 nm excitation peak through modulation of the ionized state of nearby Glu222, while the F64L mutation is responsible for the improved folding efficiency at 37 °C. Our EGFP sample is acquired from Biovision. It is expressed and purified from transformed *Escherichia coli* using a method that ensures high purity and maximal green fluorescence. The molar concentration is much less than 1 mM, above which the β-barrel protection becomes ineffective[16].

**Spontaneous four-wave mixing process in EGFP**. Figure 5a shows the relevant partial energy-level diagram of the electronic states in EGFP. The molecular system contains an electronic ground state ($S_0$), a first excited state ($S_1$), and multiple higher excited states ($S_{n>1}$) with vibronic levels on each of the electronic states. The gap between $S_1$ and $S_0$ is ~500 nm, corresponding to the fluorescence spectrum of EGFP. In our experiment, as shown in Fig. 5b, two pump photons ($\lambda_p = 2\pi c/\omega_p = 785$ nm) drive the system from the ground virtual state ($V_g$) to a final virtual state ($V_f$) through an intermediate virtual state ($V_i$). Nondegenerate signal ($\omega_s$) and idler ($\omega_i$) photons are simultaneously emitted while the system reverts back to $V_g$.

**Analytical expressions for $E_n$ and $S_L$ of $\rho'(p)$**. For a biqubit system, the entanglement of formation ($E_n$) is given as $E_n = H\left(\left(1 + \sqrt{1 - \gamma^2}\right)/2\right)$, where $\gamma$ is the

concurrence, and the function $H(x) = -x \log_2 x - (1 - x) \log_2(1 - x)$[45]. The concurrence ($\gamma$) is defined as max$\{0, \lambda_1 - \lambda_2 - \lambda_3 - \lambda_4\}$, where the $\lambda_i$s are the square roots of the eigenvalues, in a decreasing order, of $\rho(\sigma_y \otimes \sigma_y)\rho^*(\sigma_y \otimes \sigma_y)$, with $\sigma_y$ as the second Pauli matrix and $\rho^*$ as the complex conjugate of $\rho$. For $\rho'$, $\gamma = 1 - 2p$. The linear entropy is given as $S_L = 4(1 - \text{tr}(\rho^2))/3 = 8p(1 - p)/3$.

**Analytical expressions of the correlations and $R_n$ of $\rho'(p)$**. Total correlation is given as $I = -p \log_2(p/2) - (2 - p)\log_2(1 - p/2) + x_1 \log_2(x_1) + x_2 \log_2(x_2)$, where $x_{1,2} = 0.5 \pm 0.5\sqrt{\left(p^2 + (1 - p)^2\right)}$. The classical correlation is given as $C = -(p/2)\log_2(p/2) - (1 - p/2)\log_2(1 - p/2)$. The quantum correlation is given as $Q = -(p/2)\log_2(p/2) - (1 - p/2)\log_2(1 - p/2) + x_1 \log_2(x_1) + x_2 \log_2(x_2)$. The relative entropy of entanglement is given as $R_n = -(p/2)\log_2(p/2) - (1 - p/2)\log_2(1 - p/2) + x_1 \log_2(x_1) + x_2 \log_2(x_2)$.

**Data availability**. The data that support the findings of this study are available from the corresponding author on reasonable request.

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

## Acknowledgements
This work was supported in part by the DARPA QuBE program (grant no. N66001-10-1-4067).

## Author contributions
S.S., K.F.L., and P.K. designed the experiments; S.S. performed all the experiments; S.S. and K.F.L. analyzed the data; S.S. and K.F.L. prepared the manuscript; S.S., K.F.L. and P. K. discussed the results and provided comments on the manuscript. All authors reviewed the manuscript.

## Additional information

**Competing interests:** The authors declare no competing financial interests.

