## [Peer Review File · Nature Communications]

Reviewers' comments:

Reviewer #2 (Remarks to the Author):

The authors have adequately addressed my concerns. I see it being of interest to a broad audience in Nat Comms.

Reviewer #4 (Remarks to the Author):

The paper reports the creation of entangled light in a biologically active molecule GFP. This could be useful in the future for quantum light spectroscopy in biology where the quantum light is generated in situ. The results are new and the evidence for entanglement is solid. Generally the paper is well written

1 terminology. the technique should be called two photon induced parametric down conversion rather than spontaneous 4 wave mixing. 4WM usually implies three incoming classical fields that generate one quantum field. The present technique is actually PDC.

2 there is an extensive references and introduction to coherence in quantum biology. many of those results are highly controversial and exaggerated to say the least. Since the present scheme is off resonant, it can work in any similar generic multilevel system and there is nothing unusual in that GFP happens to be a biological molecule. Fast dephasing does not affect off resonant signals. The only relevant biological point is my comment #1.

All references to quantum biology and long lived coherence should be eliminated

3 what is the spectrum of W_i and W_s ? will be nice to plot it

4 the authors have to make a full analogy with type-1 and type-2 PDC. I believe it is a type-1 generation that is typically done using 2 crystals with orthogonal axes. Here instead two pumps with orthogonal polarization are used. It would be helpful to see this discussed.

5 spectroscopy is a primary future application see eg REVIEWS OF MODERN PHYSICS, VOLUME 88, OCTOBER–DECEMBER 2016 045008-1

This paper could be suitable to nat comm after a major revision

Reviewers' comments:

Reviewer #2 (Remarks to the Author):

The authors have adequately addressed my concerns. I see it being of interest to a broad audience in Nat. Commun.

Reply: We sincerely thank the reviewer for helping us to improve the clarity and quality of the manuscript.

Reviewer #4 (Remarks to the Author):

The paper reports the creation of entangled light in a biologically active molecule GFP. This could be useful in the future for quantum light spectroscopy in biology where the quantum light is generated in situ. The results are new and the evidence for entanglement is solid. Generally, the paper is well written.

1. Terminology. the technique should be called two photon induced parametric down conversion rather than spontaneous 4 wave mixing. 4WM usually implies three incoming classical fields that generate one quantum field. The present technique is actually PDC.

2. There is an extensive references and introduction to coherence in quantum biology. Many of those results are highly controversial and exaggerated to say the least. Since the present scheme is off resonant, it can work in any similar generic multilevel system and there is nothing unusual in that GFP happens to be a biological molecule. Fast dephasing does not affect off resonant

signals. The only relevant biological point is my comment #1. All references to quantum biology and long lived coherence should be eliminated.

3. What is the spectrum of ω_i and ω_s ? will be nice to plot it.

4. The authors have to make a full analogy with type-1 and type-2 PDC. I believe it is a type-1 generation that is typically done using 2 crystals with orthogonal axes. Here instead two pumps with orthogonal polarization are used. It would be helpful to see this discussed.

5. Spectroscopy is a primary future application see e.g. REVIEWS OF MODERN PHYSICS, VOLUME 88, OCTOBER–DECEMBER 2016 045008-1

This paper could be suitable to Nat. Commun. after a major revision.

Point-by-Point Response:

1. Terminology. The technique should be called two photon induced parametric down conversion rather than spontaneous 4 wave mixing. 4WM usually implies three incoming classical fields that generate one quantum field. The present technique is actually PDC.

Reply: We agree with the reviewer that the technique is a parametric process induced by two pump photons. Parametric down conversion (PDC) utilizes the χ

(2) nonlinearity where a pump photon

is converted into an entangled-photon pair in a crystal. We also agree that four-wave mixing (FWM) process usually implies three incoming classical fields and one generated field. Since the spontaneous four-wave mixing (SFWM), which utilizes the χ

(3) nonlinearity to create a photon

pair by annihilating two pump photons, has been generally accepted by the broad audience in this field (for example, [A]), we believe SFWM would be clear to the readers.

[A]. Boyd, R. W. Nonlinear optics. (Academic, 2003).

2. There is an extensive references and introduction to coherence in quantum biology. Many of those results are highly controversial and exaggerated to say the least. Since the present scheme

is off resonant, it can work in any similar generic multilevel system and there is nothing unusual in that GFP happens to be a biological molecule. Fast dephasing does not affect off resonant signals. The only relevant biological point is my comment #1. All references to quantum biology and long lived coherence should be eliminated

Reply: We thank the reviewer for pointing this out. As suggested by the reviewer, we have eliminated the potentially misleading and irrelevant references and comments related to quantum biology and long lived coherence in the revised manuscript. Moreover, we agree with the reviewer that our result is useful for developing quantum light spectroscopy in biology, which is a promising tool in the future. The major changes are listed below.

The first two paragraphs of the INTRODUCTION on page 2 are modified as follows:

Quantum spectroscopy exploits the quantum nature of light for exploring the light-matter interactions [1]. The distinctive advantages of quantum light in spectroscopy stem from improved signal-to-noise ratio owing to strong correlation between light and matter, particularly quantum entanglement [2]. Entangled light beams retain an inherent feature that the quantum state of each beam cannot be described independently of the others. Specifically, entangled photon pairs promise to enhance the precision of measurements [3, 4], as well as provide novel control knobs in nonlinear spectroscopic applications [5, 6]. Advances in quantum biology [7] have driven our attention to novel quantum spectroscopic techniques in biological materials where the entangled light is generated in situ.

Fluorescent protein (FP) has received significant attention in biomedical research (fluorescence microscopy [8], intracellular dynamics [9], reporter gene technology [10]) because of its high quantum efficiency in absorption-emission processes [11, 12] and its ability to fuse to other proteins while maintaining fluorescence [13]. The optofluidic biolaser [14], where the gain medium is enhanced green fluorescent protein (EGFP) expressed in live cells (the definition of EGFP is given in Supplementary Note 1), has succeeded in measuring subtle changes in biological molecules. The

potential application of FPs in quantum technology, however, still awaits exploration. The generation of non-classical light, such as squeezed and entangled light, in biologically produced FPs would open up the potential for quantum spectroscopy and quantum-enhanced measurements in biological systems because entanglement can provide precision that surpasses the uncertainty principle [4]. Entanglement can be generated in FPs owing to the strong optical nonlinearity [15] and the process can be quite efficient because of the protective β -barrel structure surrounding the fluorophore [16]. In addition, since the FPs strongly couple to light, even at the single-photon level, it is feasible that quantum-optics techniques for creating, manipulating, and characterizing photonic quantum states developed for use in quantum information processing (QIP), could be directly applied to FPs. By projecting the entanglement of the quantum field back to FPs, precise preparation and control of higher excited states [17-19] are feasible.

In the DISCUSSION section on page 12, we have modified the text to the following:

Moreover, the entangled photon pairs can be engineered for manipulating vibronic states in FPs through two-photon excitation. The superiority originates from the simultaneous absorption of the entangled photon pairs [19], which avoids the decay process in the intermediate states that occur when using classical light.

3. What is the spectrum of ω_i and ω_s ? will be nice to plot it.

Reply: The spectra of ω_i and ω_s are obtained from the conservation of energy $2\omega_p = \omega_i + \omega_s$ in the FWM process. In our experiment, the signal and idler spectra have flat top shape with a FWHM bandwidth of 20 nm provided by the signal and idler filters. We now plot the filter shapes of the signal and idler channels in the Supplementary Material.

4. The authors have to make a full analogy with type-1 and type-2 PDC. I believe it is a type-1 generation that is typically done using 2 crystals with orthogonal axes. Here instead two pumps with orthogonal polarization are used. It would be helpful to see this discussed.

Reply: In our experiment, two horizontally polarized pumps enter the sample at a time slot t_1 to

create the quantum amplitude for horizontally polarized photon pair. And then, two vertically polarized pumps enter the sample at a time slot t_2 to create the quantum amplitude for vertically polarized photon pair. We avoid the interaction of the orthogonally polarized pump beams in the sample because they arrive in different time slots. The superposition of the horizontally- and vertically-polarized quantum amplitudes for creating a polarization-entangled photon pair is achieved by compensating the time difference $\Delta t = t_2 - t_1$ near the detectors.

5. Spectroscopy is a primary future application see e.g. REVIEWS OF MODERN PHYSICS, VOLUME 88, OCTOBER–DECEMBER 2016 045008-1

Reply: We appreciate the reviewer for recommending this insightful paper. It's really helpful and we have cited it.